# Shape-assisted self-assembly

Joseph F. Woods [1,3], Lucía Gallego [1,3], Pauline Pfister[1], Mounir Maaloum[2], Andreas Vargas Jentzsch [2,3] &
Michel Rickhaus [1✉]

Self-assembly and molecular recognition are critical processes both in life and material sciences. They usually depend on strong, directional non-covalent interactions to gain specificity and to make long-range organization possible. Most supramolecular constructs are also at least partially governed by topography, whose role is hard to disentangle. This makes it nearly impossible to discern the potential of shape and motion in the creation of complexity. Here, we demonstrate that long-range order in supramolecular constructs can be assisted by the topography of the individual units even in the absence of highly directional interactions. Molecular units of remarkable simplicity self-assemble in solution to give single-molecule thin two-dimensional supramolecular polymers of defined boundaries. This dramatic example spotlights the critical function that topography can have in molecular assembly and paves the path to rationally designed systems of increasing sophistication.

[1] Department of Chemistry, University of Zurich, 8057 Zurich, Switzerland. [2] SAMS Research Group, University of Strasbourg, Institut Charles Sadron, CNRS, 67200 Strasbourg, France. [3] These authors contributed equally: Joseph F. Woods, Lucía Gallego, Andreas Vargas Jentzsch. ✉email: michel.rickhaus@chem.uzh.ch

Shape plays a critical role in natural molecular recognition processes such as enzyme catalysis. As early as 1894, Emil Fischer proposed with his 'Lock and Key' model that a substrate must possess a profile complementary to that of the enzymatic cleft; otherwise, there is poor association between the two[1]. Within this understanding, one of the simplest topographical elements that can allow for (molecular) recognition is curvature. The notion of curvature has an underlying and accepted role in creating order within supramolecular—in particular crystalline—systems, but is overshadowed by the driving forces for (self-)assembly[2,3]. Beyond this well-known supportive function, the role of topography as a major driving force between alike systems in supramolecular assembly is under-explored.

Over three decades ago, the first reports on supramolecular polymers[4,5] catapulted the interest in soft dynamic materials and, particularly, hydrogen-bonded linear assemblies[6–8]. Together with other highly directional non-covalent interactions[9–14] they remain the classical choice for designing self-assembling systems due to their reversible and self-healing properties[15,16]. Less directional non-covalent interactions between π-surfaces or hydrophobic contacts seldom result in assemblies[17,18]. In the cases when they do, assistance is required from another effect in tandem[19–24] because of the limited long-range control. To a certain extent, curvature has been recognized as a feature for assembly-proficient monomers. For instance, Aida and co-workers exploited a bowl-shaped corannulene for permitting the creation of dormant supramolecular species and thus the development of living-supramolecular polymerization[25,26]. Moreover, they recently have used saddle-shaped molecules to report the first example of alternating heterochiral supramolecular copolymerization[27]. Additionally, Itami demonstrated that curved nanographenes can associate to form nanofibers purely based on dipolar π–π stacking[28]. These examples offer a small glimpse of the scope that this approach towards developing functional materials has to offer.

Herein, we report that in the absence of strong non-covalent interactions, the shape of a molecular unit guides the assembly into micrometer-long stacks. Those columns then self-assemble into two-dimensional (2D) sheets of single-molecular thickness and highly defined boundaries. These nanostructures are formed by assembly in solution of a negatively curved molecule and held together primarily by dipolar π–π interactions. This simplistic approach towards 2D materials is far less common than some of the existing strategies regularly employed when approaching self-assemblies for which a representative overview is presented in Supplementary Fig. 1[13,14,17–19,23,24]. Notably, this approach is complementary to on-surface self-assembly[29] and surface nucleated thin films[30] because the nanostructures exist persistently in solution. The advantage of this generic process, which we term 'shape-assisted self-assembly' (SASA), demonstrates the significance of shape in the self-assembly process[31], provides a crucial conceptual tool and brings us closer to control and rational design of complex bottom-up assemblies in nanotechnology in general.

When one associates curvature to shape, three classifications can be envisioned molecularly: zero, positive, and negative curvatures. Flat molecules (zero curvature) are frequently observed in supramolecular polymerizations due to ease of access to these structures synthetically[32–34]. Bowl shapes (positive)[35,36], and more so saddles (negative)[28,37,38], are far less commonly used topographies for the opposite reason. Yet, to introduce order in an assembly, negatively curved systems stand out as preferred candidates to induce eclipsed stacking because transverse rigidity is increased on both molecular and macroscopic scales[27,28,39]. Asymmetry in saddle design, in

addition to replacing a uniform surface with a framework scaffold, heightens the entropic barriers further due to unequal axes of principal curvature (Fig. 1). An energetic minimum can then be reached with eclipsing negatively curved frames such that small deviations from overlap demand high energetic penalties and, consequently, enforce order. Depending on the asymmetry of the saddle, their assemblies likely present differentiated lateral interactions permitting the formation of

## Universal Design
Curvature restricts rotational and translational freedom in stacked assemblies

more freedom ↑

disc

bowl

saddle

↓ higher order

## Molecular Realization
Schematic of a versatile, non-planar macrocycle composed of four 2,2'–connected units (carbazole and pyridine → carpyridine)

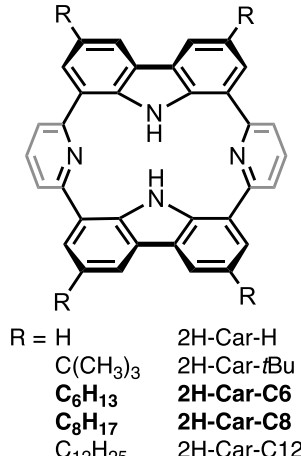

| R = H | 2H-Car-H |
| C(CH$_3$)$_3$ | 2H-Car-*t*Bu |
| **C$_6$H$_{13}$** | **2H-Car-C6** |
| **C$_8$H$_{17}$** | **2H-Car-C8** |
| C$_{12}$H$_{25}$ | 2H-Car-C12 |

## Assembly Principle
Shape-assisted self-assembly leads to formation of unidimensional stacks that subsequently laterally associate to form single-layer sheets

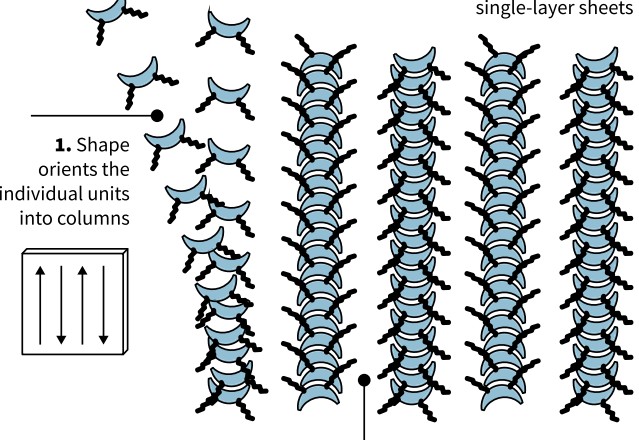

**1.** Shape orients the individual units into columns

**2.** Lateral association of the columns by weak VdW interactions

**3.** Short alkyl chains enable layer segregation

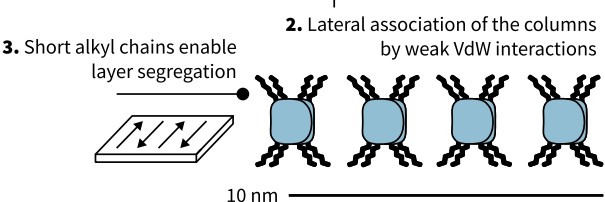

10 nm

**Fig. 1 Design principle, structural scheme, and assembly process.** Left, Shape as a design principle to restrict rotational and translational freedom. Right, Generic chemical structure of carpyridines with R group modifications to the core. Bottom, Multi-stage assembly process, where (i) units assemble into linear stacks guided by the shape of their core and driven by entropy before (ii) assembling into defined nanosheets.

tertiary structures[26]. However, the path towards these molecular topologies is not trivial: seven- or eight-membered rings are often required as the centerpiece of a saddle design but are challenging synthetic constructs to work around[27,40–43]. Five-membered rings, which would ordinarily foster bowl shapes, can be faced opposite to one another to devise 'non-classical' saddles[44]. In the absence of central polygons, this shape can also be achieved through distortions in normally planar architectures due to coordination or steric crowding[45–48]. Consideration of these design principles and the available synthetic methodologies convinced us that a contorted porphyrinoid previously reported by Müllen[44] could be tailored to become a versatile platform to explore shape-assisted linear stacking (Fig. 1).

## Results

As a first approach, peripheral alkyl substitution of four of the six available sites was proposed to (i) enhance solubility, and (ii) favor eclipsing π-surfaces. The addition of hydrogen bonding motifs was deliberately avoided because of their superior capability to order systems[27]. As reference systems, we included the unfunctionalized porphyrinoid to investigate the effect of a core without sidechains and the known macrocycle bearing *tert*-butyl groups due to its reduced propensity to self-assemble[44].

Based on Miyaura borylation of dibromocarbazoles prior to macrocyclization[44], we developed a unifying synthetic protocol. Four-fold Suzuki–Miyaura cross-coupling with a commercial pyridine gave cyclic products (up to 13% yield; 60% yield per coupling) that we name 'carpyridine' as a consequence of their interlinked constituents. Although all carpyridines could be synthesized using this route, preparation of the unsubstituted derivative **2H-Car-H** resulted in inferior yields (<1%). By subjecting **2H-Car-*t*Bu** to a reverse Friedel–Crafts alkylation reaction, we could efficiently remove the *tert*-butyl groups from the rim of the macrocycle (75% yield). Full characterization of the synthesized compounds and their intermediates is described in the Supplementary Information.

All compounds showed excellent solubility and stability in common organic solvents such as chlorinated solvents or THF. In apolar solvents like toluene or methylcyclohexane (MCH) the solubility was visibly reduced. This was corroborated by variable temperature ¹H NMR which displayed an entirely different behavior as a function of the solvent used. **2H-Car-C6** (7.5 mM) in deuterated 1,1,2,2-tetrachloroethane (TCE-$d_2$) in a temperature range from 343 to 233 K showed broadening of the aromatic signals together with an almost negligible downfield shift. However, in toluene-$d_8$ (9.2 mM), full coalescence of the pyridine NMR signals was observed at 253 K and both increasing or decreasing temperatures resulted in new environments and/or broadening (Fig. 2 and Supplementary Figs. 29–31).

Further spectroscopic evidence could be obtained from VT UV–vis and fluorescence spectroscopies (Supplementary Figs. 41 and 42). The former shows a decrease in intensity with only minor spectral changes, and the latter a decrease in fluorescence quantum yield at higher temperatures, likely due to aggregation-induced fluorescence quenching. Dynamic light scattering (DLS) also shows the presence of assembled nano objects in a 1 mM toluene solution (Supplementary Figs. 52 and 53).

Key evidence of the carpyridine ordering and SASA was obtained by microscopy. By (scanning) transmission electron microscopy (TEM; STEM) we observed extremely well-defined, micrometer-long 2D sheets when imaging samples prepared from **2H-Car-C6** (1 mM in toluene) (Fig. 3d). Sheets of larger

## Variable Temperature NMR spectra

Selected ¹H NMR resonances in toluene-$d_8$ or 1,1,2,2-tetrachloroethane-$d_2$ (TCE) and corresponding DOSY coefficients. At ~253 K in toluene-$d_8$ the system is in high exchange, while both lowering and increasing the temperatures leads to conformational rigidity

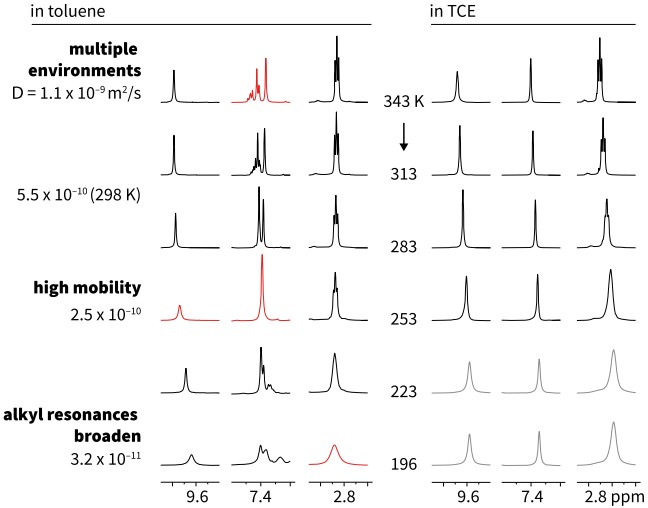

**Fig. 2 Variable temperature ¹H NMR of 2H-Car-C6 in toluene-$d_8$ and TCE-$d_2$.** Selected regions and temperatures only, see Supplementary Information for the entire datasets. In TCE broadening is only observed at low temperatures while in the aggregation-inducing solvent (toluene) distinct environments emerge at low and high temperatures. Gray signals in TCE are measured at 233 K.

dimensions were seen with **2H-Car-C8** (also 1 mM in toluene) but edge definition was consistently below that of **2H-Car-C6** (Supplementary Fig. 44). These structures were only observed for **2H-Car-C6** and **2H-Car-C8** bearing hexyl and octyl side chains, respectively, and when assembled in toluene. Longer side chains (**2H-Car-C12**), or the absence of them (**2H-Car-H**), resulted in significantly less defined aggregates (Supplementary Fig. 44). Higher resolution images and topologies of the 2D sheets from **2H-Car-C6** were obtained by atomic force microscopy (AFM) as seen in Fig. 3a. Individual sheets can be seen isolated or stacked on top of one another, with highly uniform edges allowing identification of each distinct assembly (Fig. 3b, c). In addition to single-layer sheets (Supplementary Fig. 51), holes in one multi-layer sheet allowed the measurement of the thickness of a single layer, determined to be 2 nm; a distance that is equivalent to the width of a single carpyridine molecule (Fig. 3e, f and Supplementary Fig. 60, top).

We then subjected **2H-Car-C6** TEM samples to selected area electron diffraction (SAED) on ensembles of 2D sheets due to the rapid degradation of the organic material under the electron beam. The diffraction patterns obtained (Fig. 3g, and Supplementary Fig. 49) clearly show long-range order within the structures as observed by defined diffraction spots on top of diffuse halos. We measured a principal characteristic distance of about 4 Å, which can be readily assigned to the π-π distance between the carpyridines cores as observed by crystallography and predicted by density functional theory (DFT) calculations (Fig. 4, see the "Discussion" section).

Further observations of **2H-Car-C6** in the solid-state were made with thermal analysis using differential scanning calorimetry (DSC). This showed several phase transitions and notably at least two distinct crystallization events on the cooling traces. By polarized optical microscopy, we observe at least two different (semi)crystalline phases (Supplementary Fig. 54). Notably, when the cooling rate of the melt is carefully controlled (<5 K min⁻¹),

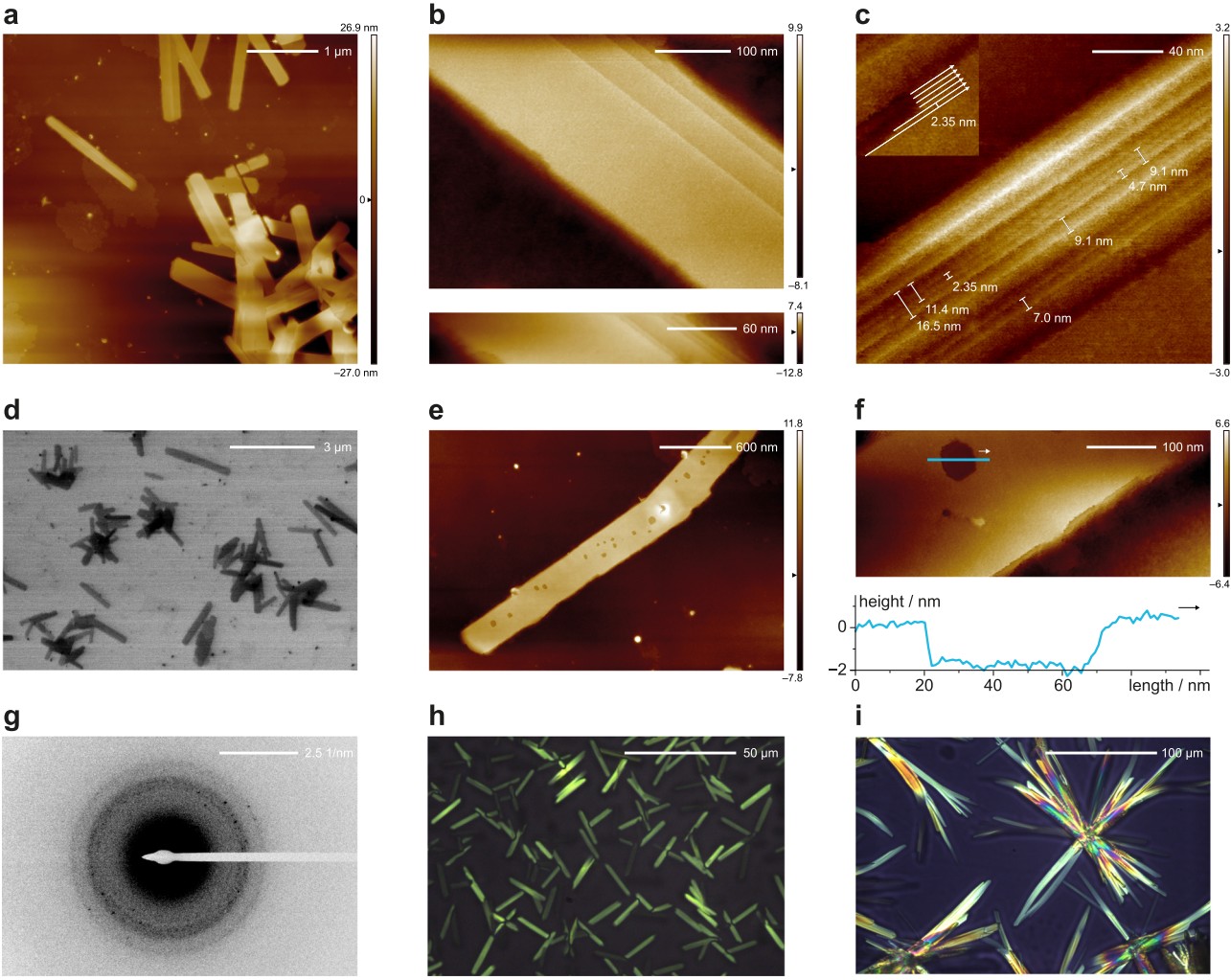

**Fig. 3 Microscopy images of 2H-Car-C6 self-assembled into 2D sheets. a** AFM images show sheets in contact with one another or isolated. **b** Sharp boundaries in multi-layer sheets allow distinction of individual units. **c** Individual columnar assemblies and their propagation direction can be distinguished in regions with defects, and by the step thickness of the edges (insert and Supplementary Fig. 50). **d** STEM image showing micrometer-long individual sheets and clusters. **e** and **f** AFM depth measurements of a hole indicate sheet thickness of 2 nm, consistent with that of a single molecule. **g** SAED pattern obtained from ensembles of 2D sheets. **h** and **i** POM images of bulk samples at different temperatures; 150 °C heating (**h**) and 25 °C after slow (<5 K min⁻¹) cooling from the isotropic melt (**i**).

the last crystallization transition yields large, highly uniform, sheet and needle-like crystalline regions of >100 μm in length, in strong support of a highly directional supramolecular interaction.

## Discussion

The results observed with **2H-Car-C6** demonstrate that it is indeed possible to obtain highly ordered, self-assembled nanosheets from monomers that do not possess strong interaction sites. Images of **2H-Car-C8** support this but also revealed a delicate balance between the shape-enhanced π–π stacking of the carpyridine core and the solvophobic interactions of the sidechains. **2H-Car-C12** demonstrates that too long sidechains prevent assemblies of significant order and that interactions at the periphery of the cycles are not the dominant driving force. On the other hand, the observations for **2H-Car-H** mean that the carpyridine core alone would assemble in a different, arguably less ordered, manner under the same conditions.

From the AFM images of **2H-Car-C6** and the clearly defined edges, we propose that the long dimension of the assembled 2D sheets is formed by ordered stacks of carpyridines, and the side edges are mainly aliphatic contacts. Such a construction reveals the critical interplay between the organizing nature of shape, locking carpyridines into 1D columns, followed by lateral van der Waals interactions resulting in highly ordered 2D assemblies[26]. The high aspect ratio (~1000:100:1) is reminiscent of crystalline sheets formed by metal–ligand interactions but the observed halo XRD patterns emphasize the soft-material nature of our nanosheets.

Additional evidence for the proposed molecular arrangement within the carpyridine stacks was obtained by single-crystal X-ray diffraction and DFT calculations. As expected, obtaining defined crystals of sufficient size in all dimensions was challenging but ultimately, single crystals of **2H-Car-C6** were obtained from a toluene/methanol mixture albeit of poor quality. Despite the heavily disordered hexyl sidechains, the obtained structures clearly show the persistent shape of the carpyridine unit and their tendency to align their principal axes of curvature in a 'slipped' stack. Although the precise arrangement found in the crystal would likely not yield ordered 2D-sheets of the observed magnitude, the relative arrangement between each two monomers is quasi-identical to the one predicted in stacked columns (see DFT

**DFT predictions on a short stack**

DFT using wB97XD/6-31(d,p) predicts two favorable conformations of **2H-Car-H** on a shortened stack, either as an alternating stack (side and front view below) or a tilt-slipped arrangement (see SI) with very similar energies $\Delta E_{DFT}$ = 2.6 kJ mol$^{-1}$.

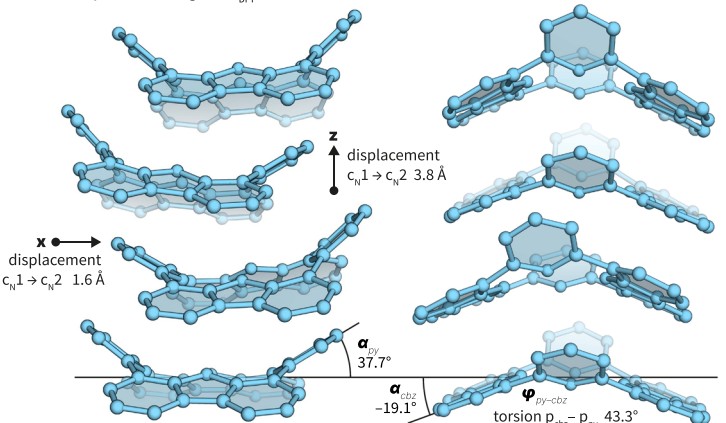

Sidechains are omitted for clarity. $c_N$ = nitrogen centroid, $p_{py}$ = plane defined by $N_{py}$ and $C_{py}$, $p_{cbz}$ = plane defined by $N_{cbz}$ and $C_{cbz}$.

**Structural parameters in the solid-state**

Part of the unit cell of **2H-Car-C6** is found in a slipped and tilted π-π arrangement, with little lateral rotation (~2° per unit). The curvatures and sidechains (disordered, omitted) of each three-unit stack are oriented isotropically. Deviations from planarity are –23.1° ($\alpha_{cbz}$) / 37.6° ($\alpha_{py}$) and 37.5° carbazole-pyridine torsion ($\varphi_{cbz-py}$).

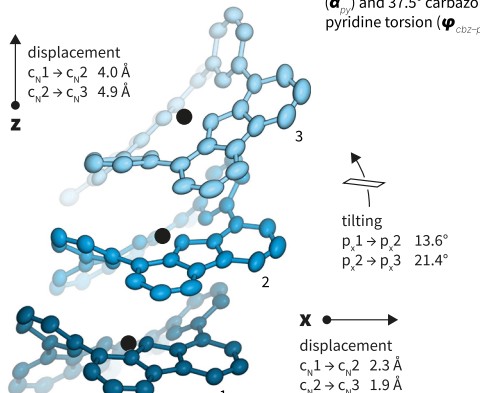

Sidechains and rest of the unit cell omitted for clarity. $c_N$ = nitrogen centroid (black dot), $p_x$ = nitrogen plane, $p_z$ = plane defined by $N_{pyridine}$ and $C_{4,pyridine}$. Thermal ellipsoids drawn at the 50% probability level.

**Fig. 4 Structural considerations.** Left, DFT calculations of a four-unit stack of the carpyridine core (**2H-Car-H**) showing the calculated arrangement with retention of the principal axis of curvature. Right, Structural parameters derived from the X-ray single crystal structure of **2H-Car-C6** showing a slipped arrangement also predicted by DFT (see Supplementary Information).

below). It further confirms that by increasing the barriers to translation and rotation of stack formation, the curvature of the saddle-shaped molecules enhances the order of the assembly as postulated.

Our theoretical calculations by DFT using various functionals (B3LYP, wB97XD, PBE0 and the 6–31(d,p) basis set) predict that the preferred assembled configuration of the carpyridine core is an alternated stacked conformation (Supplementary Figs. 57–59). The displaced saddle conformation as observed in the crystal structure is another local minimum but at a slightly higher energy ($\Delta E_{DFT}$ ~ 3 kJ mol$^{-1}$ in average among all functionals) in agreement with the polymorphism we here report. Although saddles are translationally displaced by 1.6 Å from centroid to centroid, all carpyridines retain the same orientation along the principal axis of the stacks. In both arrangements, the predicted inter-carpyridine distance between centroids is 3.8 Å and within the expected range of π–π stacking effects[49,50]. Our DFT studies were complemented by semi-empirical calculations using the PM6 method, showing that the addition of the hexyl chains (stacks of six **2H-Car-C6** units were studied) leads to the preferred stable linear alternated stacked structure proposed above (Supplementary Fig. 60).

Our studies in solution demonstrate the presence of supramolecular aggregates as seen in both NMR and DLS. Although broadening of resonances in $^1$H NMR studies at lower temperatures is expected for classical supramolecular polymers, broadening or splitting at higher temperatures has been attributed to entropy-driven processes[51,52]. Together with the relatively invariant spectral response to temperature this suggests a possible shape-assisted process between complementary curvatures[53], whereby toluene is displaced from inter-carpyridine surfaces, which further highlights the previously reported key role of solvent in any self-assembly process.

The role of shape in supramolecular polymerization which we describe herein can be found concealed in countless previous reports. In most hydrogen-bonded systems, for instance, hydrogen-bond donors and acceptors need to arrange in precise 3D configurations to permit assembly, hence restricting the barriers to translation and rotation of stacks by the formation of strong interactions. It is also well-recognized in crystal engineering where packing effects largely drive the formation of crystalline material, notably in absence of hydrogen bonds or columbic interactions. Nevertheless, molecular shape is rarely used as a design principle in supramolecular polymerization and we have not found any existing example where non-crystalline structures can show the degree of long-range order that we report without relying on classical highly directional non-covalent interactions.

Overall, the simplicity of the carpyridine supramolecular units, which results in the self-assembly of large 2D sheets of single molecular thickness, showcases the potential of shape-enhanced effects. It is only possible because its topological elements enforce shape-assisted directional assembly—remarkably achieved in absence of strong and directional interactions—but also enough flexibility to prevent the system from collapsing in solution like most rigid 2D materials. The 2D sheets themselves can only be assembled as a consequence of the delicate equilibrium between entropic gains and the competition of both dipolar π–π and hydrophobic interactions. Due to its simplicity, the example we present here is uniquely helpful to illustrate the critical role of shape in the self-assembly of molecular nano- and microstructures. It further suggests that the significance of shape might have been underestimated in the past. As the obtained assembly is a direct consequence of the underlying topography, SASA holds substantial promise as a design concept for new soft-matter materials and its implementations in nanotechnological applications such as sensing and nanofabrication.

## Methods

**General information**. All reagents and solvents were purchased from Avantor, Chemie Brunschwig AG, Sigma-Aldrich or Thermo Fisher and used without further purification. Dry solvents were obtained using a solvent purification system (Pure Solv PS-MD-4EN, Innovative Technology Inc.) equipped with alumina drying columns under argon. Reaction control was performed using analytical thin layer chromatography (TLC) on aluminum sheets coated with silica gel 60 F254 (Merck) or gas chromatography–mass spectrometry (GCMS) (Shimadzu Gas Chromatograph GC-2010 Plus, GCMS-QP2010 SE). Visualization of TLC plates was achieved using UV light at 254 or 366 nm. Flash column chromatography (FC) was performed using SiO$_2$ (60 Å, 230–400 mesh, particle size 0.063–0.200 mm).

Size-exclusion chromatography (SEC) was performed with Purolite Gel Poly-styrene beads crosslinked with 1% divinylbenzene using chloroform as an eluent. Preparative gel permeation chromatography (GPC) was carried out on a Shimadzu recycling GPC system equipped with an LC-20AR prominence liquid chromato-graph pump, an SPDM40 photodiode array detector, a DGU-403 degassing and a CBM-40 system controller using two ReproGel 500 GPC columns (5 μm, 20 × 600 mm) with chloroform as the eluent passing through at a rate of 3.5 mL per minute.

All *NMR spectra* were recorded on AV2-400, AV2-500, or AV2-600 MHz Bruker spectrometers at 298 K unless stated otherwise. Chemical shifts are given in ppm and the spectra are calibrated using the residual chloroform signals (7.26 ppm for [1]H NMR and 77.16 ppm for [13]C NMR), the residual dimethylsulfoxide signals (2.50 ppm for [1]H NMR and 39.52 ppm for [13]C NMR), the residual toluene signals (7.09, 7.00, 6.98 and 2.09 ppm for [1]H NMR) and the residual 1,1,2,2-tetrachloroethane signals (6.00 ppm for [1]H NMR). Coupling constants *J* are given in Hz and multiplicities are abbreviated as follows: s (singlet), d (doublet), t (triplet), m (multiplet), br (broad). [1]H NMR assignments were made using 2D NMR methods (COSY, NOESY, HSQC, HMBC). DOSY measurements allowed for calculation of the apparent MWs following a reported excel sheet from the University of Manchester[54].

*High resolution mass spectra (HR-MS)* were recorded by the Mass Spectrometric Service at the University of Zurich on a Dionex Ultimate 3000 UHPLC system (ThermoFischer Scientifics, Germering, Germany) connected to a QExactive MS with a heated ESI source (ThermoFisher Scientific, Bremen, Germany) or on a double-focusing (BE geometry) magnetic sector mass spectrometer DFS (ThermoFisher Scientific, Bremen, Germany) with a heated EI source.

*UV-vis absorbance measurements* were recorded at 298 K or at the given temperature with a Shimadzu UV–Visible Spectrophotometer UV-1900 Series using quartz 1 cm or 1 mm cuvettes.

*Fluorescence measurements* were carried out using a calibrated Edinburgh Instruments FS5 spectrofluorometer equipped with an SC-25 temperature-controlled holder TE-Cooled-Standard cell for emission spectra and an SC-30-Integrating Sphere cell for obtaining quantum yields. All solvents used for spectrophotometric analysis were of analytical grade.

*Dynamic light scattering (DLS)* was measured using a Malvern-Nano-ZS90 instrument.

*Differential scanning calorimetry (DSC)* was measured using a DSC 2500 instrument.

*IR spectra* were recorded in a SpectrumTwo FT-IR spectrometer (Perkin–Elmer) equipped with a Specac Golden GateTM ATR (attenuated total reflection) accessory.

**Microscopy**. *TEM imaging* was performed using either a CM12 Philips microscope equipped with an MVIII (Soft Imaging System) CCD camera or an FEI Tecnai G2 Spirit microscope using a side-mounted digital camera Gatan Orius 1000. Samples were analyzed in Bright Field Mode with an LaB6 cathode and 120 kV tension. Image treatments were performed by using analySIS (Soft Imaging System) software.

*TEM diffraction* patterns were obtained with support of the Center for Microscopy and Image Analysis, University of Zurich using an FEI Tecnai G2 Spirit microscope working at a 120 kV tension using a side-mounted digital camera Gatan Orius 1000 (4k × 2.6k pixels). 14 bits. Pixel size 9 μm. Frame rate up to 14 fps at binning 4.

*SEM and STEM imaging* were performed on an FEG-SEM Hitachi SU8010 at room temperature at 20 kV.

*Polarized optical microscopy (POM)* images were acquired using a Nikon Labophot-2 microscope with an integrated Infinity3 camera.

*AFM images* were obtained as described before (https://doi.org/10.1002/chem.201902404) by scanning the samples using a Nanoscope 8 (Bruker) operated in Peak-Force tapping mode. Peak-Force AFM is based on Peak force tapping technology, during which the probe is oscillated in a similar fashion as it is in tapping mode, but at a frequency far below the resonance frequency. Each time the tip and the sample are brought together, a force curve is captured. These forces can be controlled at levels much lower than contact mode and even lower than tapping mode allowing operation on even the most delicate soft samples, as it is the case here. An ultra-sharp silicon tip on a nitride lever were used (Bruker, Scanasyst with spring constant of 0.4 N/m). During AFM imaging, the force was reduced in order to avoid dragging molecules by the tip. All analyses of the images were conducted in the integrated software.

**Sample preparation**. *UV-vis absorbance measurements* were carried out with ca. $10^{-5}$ M solutions in the solvent specified using a 1 cm quartz cuvette. *Variable temperature (VT) UV–vis absorbance measurements* of **2H-Car-C6** were performed on a 1 mM solution in toluene using a 1 mm quartz cuvette. *Emission spectra and fluorescence quantum yields* were measured on samples with an optical density of 0.06–0.10 using a 1 cm quartz cuvette. *Solid-state fluorescence* of **2H-Car-C6** was measured after deposition of a 1 mM solution in toluene on a glass microscope slide and evaporation of the solvent. *DLS* was measured on a 1 mM solution in toluene in a 1 cm quartz cuvette. *DSC* was measured on 1.663 mg of **2H-Car-C6**

after deposition of the sample in the crucible from a solution in toluene and evaporation of the organic solvent. *IR* was measured in thin film.

The samples for *electron microscopy* were prepared at a typical concentration of 1 mM solution in toluene: a 1 mM solution in toluene (0.2–1.0 mL) in a sealed vial was heated to about 100 °C for a few seconds and was allowed to regain room temperature. Through the process, the vials were kept at a uniform temperature and concentration by slow convexing movements. After 10–15 min at room temperature, 5 μL of the formed dispersion were deposited on a C/Cu or C/Ni grid. The solution was immediately bloated off the grid with a filter paper. After full evaporation of the solvent, the samples were used for *TEM, SEM, STEM and AFM imaging.*

The samples for *polarized optical microscopy* were prepared as described above. Once the vial was left for 10–15 min at room temperature, 5 μL of the formed dispersion were deposited on a glass microscope slide and the solvent was evaporated. *VT polarized optical microscopy* was performed after heating the glass microscope slide to a certain temperature using an IKA RCT basic hotplate. The newly formed structures showed to be stable while performing the imaging.

**X-ray diffraction**. A crystal of $C_{58}H_{68}N_4$, obtained from toluene and MeOH, was mounted on a cryo-loop and used for a low-temperature X-ray structure determination. All measurements were made on a *Rigaku Oxford Diffraction XtaLAB Synergy* diffractometer[55] with a *Pilatus 200 K* hybrid pixel area detector using Cu K$\alpha$ radiation ($\lambda = 1.54184$ Å) from a *PhotonJet* micro-focus X-ray source and an *Oxford Cryosystems Cryostream 800* cooler. Data reduction was performed with *CrysAlisPro*[55]. The intensities were corrected for Lorentz and polarization effects, and a numerical absorption correction[56] was applied. The space group was uniquely determined by the systematic absences. Equivalent reflections were merged. The structure was solved by dual space methods using *SHELXT-2018*[57], which revealed the positions of all non-hydrogen atoms. Neutral atom scattering factors for non-hydrogen atoms were taken from Maslen et al.[58], and the scattering factors for H-atoms were taken from Stewart et al.[59].

**Computational analysis**. All molecular models were constructed using the GaussView software[60]. Both DFT and semi-empirical calculations were performed using the Gaussian 16 revision C.01 software[61]. Three different functionals, wB97XD[62], PBE0[63] and B3LYP-D3[64,65], using the 6–31G(d,p) basis set were chosen to avoid functional-specific errors. In all cases, the absence of imaginary frequencies (for ground-state calculations) or the presence of a unique imaginary frequency (for transition state calculations) was confirmed. The PM6[66] method as implemented in Gaussian was employed for the semi-empirical calculations. The cartesian coordinates of the compounds and assembled structures are provided as separated *xyz* files.

## Data availability

All data are available either in the main text, the supplementary materials or by request of the corresponding author.

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

## Acknowledgements

Imaging was performed with support of the Center for Microscopy and Image Analysis, University of Zurich, and the microscopy facility at the ICS, Strasbourg. We also thank S. Jurt, T. Fox and N. Bross for assistance with NMR measurements, the Mass Spectrometry Laboratory at the University of Zurich for MS measurements, K. Feldman for DSC measurements, G. Patzke for DLS measurements, P. Zwick and A. Wildi for helpful discussions and K. Gademann for generously hosting and supporting our research group. M.R. gratefully acknowledges funding from the Swiss National Science Foundation grant PZ00P2_180101. J.F.W. thanks the University of Zurich Forschungskredit (FK-21-107) for support.

## Author contributions

J.F.W., L.G., A.V.J., and M.R. conceived, analyzed, validated, and visualized the work herein presented. J.F.W., L.G., P.P., M.M., A.V.J., and M.R. carried out the investigation

and developed the methodology. Funding was acquired by J.F.W. and M.R. and resources by A.V.J. and M.R. M.R. supervised the work. J.F.W., L.G., A.V.J. and M.R. wrote the manuscript.

## Competing interests

The authors declare no competing interests.
