## [Peer Review File · Nature Communications]

Reviewers' Comments:

Reviewer #1:

Remarks to the Author:

The authors have addressed a large number, but not all of the issues raised by the reviewers of their initial submission. I had previously been supportive of a transfer to NatComm and I generally maintain this view. Arguably, my most important criticism was related to the effect that solvophobic interactions play beyond shape in the observed (hierarchical) self-assembly. The authors have now addressed this issue somewhat in the discussion. However, a systematic study of side chains would be ideally suited to clarify the issue, at least on a preliminary level. So, I was very surprised to read in the rebuttal letter that such a systematic study has actually been carried out ("with variation in the alkyl chain between C2 and C8 – shows that these systems too adopt 2D sheet structures, although of different qualities."), but the data is not included in the current manuscript for reasons that are frankly not convincing ("These data are not intended to be a part of this publication as it separates from our conceptual demonstration but show that our conclusions are very strong.")

I strongly believe that this present paper has to speak for itself and it really requires at least one more example of a side chain giving rise to (hierarchical) self-assembly. Only then, the reader can roughly assess the generality of the findings and decide how to make sense of the role of shape and solvophobicity without having to wait for the publication of a follow-up paper.

Nature Communications manuscript NCOMMS-22-19408-T

Title: Shape-Assisted Self-Assembly

Author for correspondence: Michel Rickhaus

Reply to Referee's Comments and Details of Revisions

Referee #1 (Remarks to the Author):

R1 The authors have addressed a large number, but not all of the issues raised by the reviewers of their initial submission. I had previously been supportive of a transfer to NatComm and I generally maintain this view.

Reply: We are thankful to Referee # 1 for their continued support of the manuscript and the improvements that resulted from their prior comments.

R1 Arguably, my most important criticism was related to the effect that solvophobic interactions play beyond shape in the observed (hierarchical) self-assembly. The authors have now addressed this issue somewhat in the discussion. However, a systematic study of side chains would be ideally suited to clarify the issue, at least on a preliminary level. So, I was very surprised to read in the rebuttal letter that such a systematic study has actually been carried out ("with variation in the alkyl chain between C2 and C8 – shows that these systems too adopt 2D sheet structures, although of different qualities."), but the data is not included in the current manuscript for reasons that are frankly not convincing ("These data are not intended to be a part of this publication as it separates from our conceptional demonstration but show that our conclusions are very strong."). I strongly believe that this present paper has to speak for itself and it really requires at least one more example of a side chain giving rise to (hierarchical) self-assembly. Only then, the reader can roughly assess the generality of the findings and decide how to make sense of the role of shape and solvophobicity without having to wait for the publication of a follow-up paper.

Reply: Over the course of the assessment of this manuscript first in Nature and now here, we have indeed continued our studies to gain a better understanding of the reported system. We agree that expanding the scope to another example can greatly underline the generality of the approach while remaining in the (in our opinion) suitable format of a communication. The figure below shows the preliminary TEM images that we could gather from the derivatives with sidechain lengths of 0, 2, 4, 5, 6, 8 and 12 carbons, respectively. We can immediately spot that elongation of the chain from C6 to C8 still results in assembly, although of poorer quality. That underlines the fact that the sidechains aid in layer segregation, but impact the delicate nature of the assembly. Consequently, the assembly breaks down with long sidechains (C12). For that reason, we have decided to include this example in the manuscript, including full synthetic protocols, spectroscopy and TEM images. We have indicated the changes in the manuscript and the ESI accordingly.

We stand by our decision to not include chains shorter than C6 (with the exception of C0) into the present manuscript. One important reason for this decision is that most of these compounds have only been successfully synthesized in recent weeks and require a careful study to ensure the quality of the results and interpretations that will be presented in due time. At this stage, from our preliminary data we see that indeed C2-C5 are forming assemblies – the shape assisted principle holds as we believe them to still aggregate as columns (secondary structure). The formed *tertiary structures* are very strongly responsive to the employed sidechain. While C5 still forms the sheets as presented in this paper (and potentially even of better quality), C4 leads to what we believe to be folded sheets, while further shortening of the chain largely impacts the x:y ratio to give squared nanosheets. In the absence of sidechain-bias (C0) the assembly likely collapses into aggregations of columns. While this hints at a quite rational and delicate control of the obtained tertiary structure, we believe that this is beyond the scope of the current manuscript and still requires substantial investigations (primarily column orientation, solvent dependence, AFM profiles, assembly mechanism, electrical and optical properties).

Preliminary Observations

2H-Car-H

No significant assemblies observed

2H-Car-C2

Predominantly squared sheets
Aspect ratio ~ 1500:1000:1-10 (*l:w:h*)
Lower directional bias than C6
High edge definition

2H-Car-C4

Predominantly folded sheets alongside
clusters of smaller sheets
~1000:N/A:1-10

2H-Car-C5

Well defined rectangular sheets
with occasional folding
~5000:1000:1-10

2H-Car-C6

Rectangular sheets
Confirmed single layers
Defined edges
~1500:500:1

2H-Car-C8

Rectangular sheets
Less defined assemblies
~2000:500:1

2H-Car-C12

No significant assemblies observed

Reviewers' Comments:

Reviewer #1:

Remarks to the Author:

The authors have included another example (C8 chain), which will help the reader to assess the generality of the findings and decide how to make sense of the role of shape and the solvophobic effect (in my view both play a role and the follow-up study will clarify things further). I recommend acceptance without delay.